# TdmTracker: Multi-Object Tracker Guided by Trajectory Distribution Map

Yuxuan Gao [ID], Xiaohui Gu, Qiang Gao, Runmin Hou and Yuanlong Hou *

School of Mechanical Engineering, Nanjing University of Science and Technology, Nanjing 210094, China; gaoyuxuan1994@njust.edu.cn (Y.G.); gxiaohui@njust.edu.cn (X.G.); gaoq0916@sina.com (Q.G.); hourunmin1102@njust.edu.cn (R.H.)
* Correspondence: houyuanlong1964@163.com

**Abstract:** With the great progress of object detection, some detection-based multiple object tracking (MOT) paradigms begin to emerge, including tracking-by-detection, joint detection and tracking, and attention mechanism-based MOT. Due to the separately executed detection, embedding, and data association, tracking-by-detection-based methods are much less efficient than other end-to-end MOT methods. Therefore, recent works are devoted to integrating these separate processes into an end-to-end paradigm. Some of the transformer-based end-to-end methods introducing track queries to detect targets have achieved good results. Self-attention and track query of these methods has given us some inspiration. Moreover, we adopt optimized class query instead of static learned object query to detect new-coming objects of target category. In this work, we present a novel anchor-free attention mechanism-based end-to-end model TdmTracker, where we propose a trajectory distribution map to guide position prediction, and introduce an adaptive query embedding set and query-key attention mechanism to detect tracked objects in the current frame. The experimental results on MOT17 dataset show that the TdmTracker achieves a good speed-accuracy trade-off compared with other state-of-the-arts.

**Keywords:** multiple object tracking; end-to-end method; attention mechanism; adaptive query embedding set; trajectory distribution map

## 1. Introduction

Object tracking is an essential component for computer vision. In particular, multi-object tracking (MOT) attracts much attention owing to its strong practical merit. MOT aims to continuously locate multiple targets in video frames as their trajectories and label different targets with different track identities. The recent progress in deep learning has led to great improvement of object detection performance, which makes tracking-by-detection a popular paradigm in MOT [1]. Some advanced image retrieval [2,3] and preprocessing [4] technologies have also been proposed to improve the accuracy of computer vision models. However, MOT methods of the tracking-by-detection paradigm are limited by complex pipeline, which brings a lot of computational cost and makes them not real-time. Furthermore, the disordered target pairs between two consecutive frames and the incomplete detection in each frame bring great challenges to the tracking algorithm.

In the tracking-by-detection paradigm, the output of object detection is the input of the tracking algorithm, so the performance of the detection algorithm will seriously affect the effect of the whole tracking method. Afterward, the subsequent problem is to correctly associate the targets between frames by calculating similarity between features extracted from detection patches in the front and back frames and considering the location of the tracked objects. Traditional tracking-by-detection paradigm treats MOT as two tasks, object detection and data association [5]. In terms of object detection, there are many mature backbone networks for feature extraction such as VGG16, GoogLeNet, ResNet, DenseNet,

Darnet19, etc. Most state-of-the-art models, such as Faster R-CNN, YOLO, SSD, DSOD, etc., use the features extracted by these backbone networks for object classification and detection. The maturity of object detection technology makes current MOT research focus on optimizing detection, feature extraction, trajectory prediction, and data association [1,6]. SORT [6] and DeepSORT [1] are the classical methods of the tracking-by-detection paradigm. In these two methods, the detection patches of the target object are first obtained by an off-the-shelf detector, then the cost matrix is generated according to the appearance feature similarity and motion affinity of the detected patches, and finally the matching algorithm is used to match the same targets in the front and back frames so as to achieve multi-object tracking. Extracting features to calculate similarity significantly improves the accuracy of the model; hence, most recently, tracking-by-detection methods contain three sequential subtasks: object detection, feature extraction, and data association. In addition to the features extracted by deep network, some handcrafted features for specialized recognition have been used in recent work [7–9]. A survey of extant studies on gait recognition [7] finds that the information of gait is usually obtained from different parts of silhouettes. Reference [8] used handcrafted features based on Oriented Fast and Rotated BRIEF (Binary Robust Independent Elementary Features) and Scale Invariant Feature Transform (SIFT) features to realize efficient object recognition. In [9], speeded up robust features (SURF) and SIFT are used for feature extraction to implement a face recognition method. However, splitting the whole task into isolated subtasks may lead to local optima and much computation cost. Considering the efficiency of the tracking model, more and more works focus on the whole tracking speed to optimize the model architecture. Therefore, the emergence of joint detection and tracking paradigm has become an important stride in the development of MOT. Since JDE [10] incorporated the appearance embedding model into a single-shot detector, which avoided re-computation by sharing the same set of features, researchers have tended to solve object detection, feature extraction, and data association in a whole network, namely an end-to-end solution, such as FairMOT [11], CenterTrack [12], and CTracker [13]. These methods are more real-time than tracking-by-detection methods, and we are inspired to implement MOT with an end-to-end solution to realize higher practical merit. Hence, our model directly extracts the deep learning features of the whole picture, realizes feature matching through attention mechanism, and combines motion state by weighting the predicted trajectory distribution map of the current frame. Through the identification of different tracking targets on different channels, re-identification of targets is realized. It avoids recalculating features and integrates object detection, feature extraction, and data association into a unified lightweight model.

Query-key attention mechanism also benefits our model. As suggested in recent works [14–17], attention mechanism has great prospect in computer vision. DETR [14] applied the transformer widely used in the NLP field to the field of object detection and achieved good results. Reference [15] proposed a general-purpose few-shot object detector, through the well-designed Attention-RPN, Multi-Relation Detector, and contrastive training strategy, and the network can squeeze out the matching relationship between targets by training on a high-diversity dataset FSOD and carry out reliable detection of novel categories without fine-tuning. It inspires us to train the model to learn a general matching relationship to distinguish objects of the same category from those of different categories instead of learning the details of each category separately. This enables the model to have better generalization ability for novel categories. In particular, as MOT methods, TransTrack [18] and TrackFormer [16] introduce the transformer to encode frame feature and queries in the self-attention manner and use feature embedding of previous detection objects as query for searching tracked targets in the current frame and static learned object query [14] for detecting new-coming objects. We learn from their idea of using self-attention to encode the relationship between pixel patches and the relationship between queries to obtain more representative embedding. Moreover, we argue that these end-to-end models only use the IoU of the objects detected in the front and back frames for position matching, which does not make full use of the motion information of the tracked targets. Instead of

IoU matching, we proposed a trajectory distribution map to introduce historical position information of tracked objects to predict the position in the current frame so as to improve the accuracy of position matching.

In our work, we present a novel anchor-free attention mechanism-based end-to-end MOT model, referred to as TdmTracker. Considering that most methods based on attention mechanism use static learned object query to detect new-coming objects, causing many foreground objects of other categories to be detected in complex background; thus, our model adopts a comparative learning mode similar to that in few-shot learning and utilizes optimized class query to detect new-coming objects of target category. To be specific, we refer to the structure of the few-shot detection model proposed in Reference [19] to a certain extent for introducing query-key attention mechanism in a one-stage anchor-free manner, and our model takes optimized class query and track query to form an adaptive query embedding set to realize the unification of query-key attention operations. In addition, we introduce self-attention to produce a good representation for feature embedding. In order to integrate data association into the end-to-end tracking process and make full use of the historical position information of the tracked object and the corresponding timing information, we propose a trajectory distribution map for motion prediction to realize tracking conditioned detection and association through predicted trajectory.

The contributions of our work are summarized as follows:

- We propose TdmTracker, a novel anchor-free attention mechanism-based end-to-end MOT model, which presents a more unified and lightweight architecture by integrating separate subtasks into a whole network. It has comparable tracking accuracy with other state-of-the-art MOT methods and achieves a relatively high efficiency.
- We adopt a comparative learning mode similar to that in few-shot learning and utilize optimized class query instead of static learned object query to detect objects of target category. Our model takes optimized class query and track query to form the adaptive query embedding set to realize the unification of query-key attention operations.
- We propose a trajectory distribution map for motion prediction to realize tracking conditioned detection and association through predicted trajectory.

## 2. Related Works

Different from single object tracking, MOT is mainly about how to solve data association problems under the premise of ID-available targets. Data association has now developed a variety of different solutions, starting from the simplest matching with IOU [20] to the popular Greedy matching algorithm [21] and Hungarian algorithm [22]. In recent years, due to the tremendous strides in object detection, most modern MOT trackers follow the tracking-by-detection paradigm. At the same time, recently, there have been methods based on joint detection and tracking and attention mechanism, which began to attract the attention of researchers. Here, we introduce three classic paradigms of MOT: tracking-by-detection, joint detection and tracking, and attention mechanism paradigms.

**Tracking-by-detection**. Within this framework, an off-the-shelf detector is first utilized to capture the target objects in video frames. After extracting appearance feature for these detection bounding boxes and obtaining motion model, the cost matrix is calculated by appearance similarity and motion affinity. Then, a matching algorithm is performed to solve the data association problem in order to form the tracklets. Reference [6] proposed the SORT method, which used Faster Region CNN (FrRCNN) for detection, Kalman filter for predicting the motion state, and the Hungarian algorithm based on the detection locations and IOU for data association, making the algorithm highly efficient. In Deep SORT [1], Mahalanobis distance is used as motion metric for evaluating the difference between the predicted Kalman state and the detection location in the current frame, and CNN is used to extract appearance feature of detection bounding boxes to calculate cosine distance with a feature gallery for each track. Furthermore, the matching cascade method is utilized to improve matching accuracy. The improved effect of Deep SORT is obvious, which greatly reduces the ID switches in SORT. Recently, some success has been achieved in

computer vision by introducing graph neural networks (GNN), which is more effective for modeling structured data. In [23,24], data association is formulated as a graph optimization problem by treating each detection as a graph node, achieving state-of-the-art performance. However, generally speaking, tracking-by-detection methods are two-step methods, which conduct object detection and appearance feature extraction separately and hence are computationally expensive [25].

**MOT based on joint detection and tracking**. In order to build a real-time MOT system, JDE [10] incorporates the appearance embedding model into a single-shot detector, which avoids re-computation by sharing the same set of features to save computation. It reports a real-time MOT system with a speed much faster than two-step methods and a tracking accuracy comparable to the state-of-the-arts. However, the tracking accuracy of the one-shot method is often lower than that of the two-step method. Reference [11] finds that this is because the learning feature embedding is not optimal, which leads to many identity switches. They also find a better way is to extract features at the estimated object centers. Because features extracted at coarse anchors may not be aligned with object centers, FairMOT [11] uses the anchor-free method for object detection and identity embedding, which can significantly improve the tracking accuracy on all benchmarks. CenterTrack [12] adopts CenterNet [26] to localize object centers and adds four additional input channels and two output channels in order to obtain the offset by comparing with the heatmap of the prior frame. Afterward, with good offset prediction, greedy matching algorithm is leveraged to associate objects across time. Reference [13] proposes an online end-to-end MOT model CTracker which first unifies object detection, feature extraction, and data association into a single end-to-end solution. This framework is also the first to convert a data association problem to a pair-wise object detection problem. In addition, they design a joint attention module to highlight informative regions for box pair regression, which further improves the performance of CTracker.

**Attention mechanism-based MOT**. With the development of attention mechanism in the field of computer vision, many state-of-the-art models based on attention mechanism have emerged. Attention mechanism helps the model to be more focused, avoiding the distraction by irrelevant yet confusing information. DETR [14] is the first successful attempt, which applies the transformer widely used in the NLP field to the field of object detection and has achieved good results. At the same time, it simplifies the NMS and anchor mechanisms commonly used in object detection and it detects the objects in the image through the learned object queries. The experimental result obtained on the MS COCO dataset is equivalent to that of Faster-RCNN [27]. Afterward, TransTrack [18] introduces the transformer architecture, which is an attention-based query-key mechanism. It extracts object features from the previous frame as a query of the current frame and utilizes a set of learned object queries for detecting new-coming objects. TransTrack conducts object detection and data association in a single-shot, simplifying the complex multi-step process. Similar to TransTrack [18], TrackFormer [16] also utilizes an attention mechanism-based encoder-decoder architecture to query objects in frames. The difference is that it proposes the scheme of using different track queries between different frames, where new-coming objects are detected by static object queries as in [14,17] and subsequently transformed to future track queries, that is, the method of combining learned object query and track feature. Thereby, it achieves detection and data association jointly in a tracking-by-attention paradigm.

## 3. TdmTracker

In this section, we present the architecture of our end-to-end multi-object tracking (MOT) model TdmTracker and the tracking pipeline. Afterward, we describe the technical details of core components of TdmTracker.

### 3.1. Architecture and Pipeline

In order to detect a complete set of objects of target category and correctly output sorted tracked objects online, TdmTracker uses queries from two sources to obtain the adaptive query embedding set. On the one hand, similar to [19], adaptive query embedding set takes optimized class queries for target categories. On the other hand, track queries, which are feature embeddings from previously detected objects, are introduced to form the adaptive query embedding set.

As shown in Figure 1, TdmTracker takes current frame, adaptive query embedding set, and a trajectory distribution map of the previous frame as inputs. The current frame is first fed to a weight shared network which is the same as the network used to extract feature embedding of the target category. Because it is mentioned in [11] that ResNet-34 [28] has fewer parameters than ResNet-50 [28] but achieves better results, we decide to utilize ResNet-34 as our weight shared network. Afterward, three feature maps are generated by top-down connecting three extracted layers from the backbone ResNet-34. Moreover, we introduce self-attention after each feature map to encode the relationship between different objects and the relationship between foreground and background. The resulting feature maps P1, P2, and P3 constitute a feature pyramid network (FPN) [29] so that TdmTracker can detect objects from three different scales.

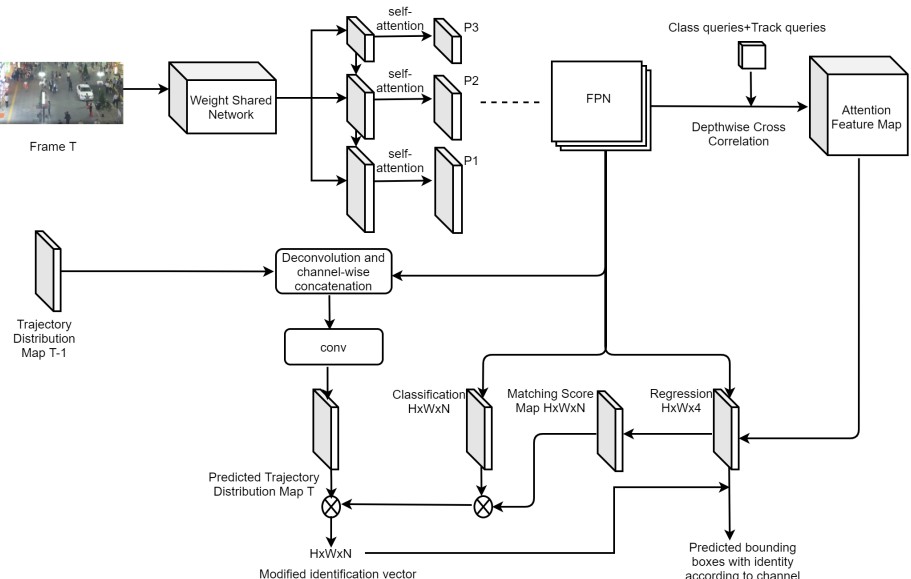

**Figure 1.** Network architecture of TdmTracker. After the current frame is input to the network, an FPN is obtained first. Given the optimized class queries, track queries in adaptive query embedding set, through query-key attention mechanism, a classification result for the objects corresponding to each query is obtained. On the other hand, trajectory distribution map of previous frame is input to the network for the branch of predicting trajectory distribution map of current frame, which guides position prediction for each tracked object and produces modified identification vector. This modified identification vector in turn guides the post-processing process of the prediction bounding boxes.

Through the branches of classification and regression bounding boxes, $C_t \in R^{H \times W \times N}$ and $R_t \in R^{H \times W \times 4}$ are obtained, respectively. $N$ is the maximum number of objects allowed to be detected, which equals the total number of queries in the adaptive query embedding set. As our model is designed to extract the feature maps according to the general rules of foreground objects, a classification $C^{H \times W \times 1}$ distinguishing pixels between foreground and background is first obtained, and then it is repeated on the channel $N$ times to form $C_t$. $R_t \in R^{H \times W \times 4}$ represents the regression bounding boxes (offsets of left, top, right, and bottom) of foreground objects. Afterward, we utilize the adaptive query embedding set composed of optimized class queries and track queries obtained from

the detection result of the previous frame to determine whether the regression bounding boxes belong to new-coming objects or tracked objects. To be specific, the similarity between each query embedding in the adaptive query embedding set and the feature map of the current frame is calculated by depth-wise cross correlation to generate an attention feature map. A matching score map is obtained from the attention feature map by referring to the results of the regression bounding boxes. The matching score map indicates the probability of each pixel in the regression bounding boxes belonging to the object corresponding to each query by calculating the average value of pixel similarity in the regression bounding box. Afterward, this matching score map performs element-wise product operation with the prior classification $C_t$. Here, we obtain a corrected classification result for the objects corresponding to the adaptive query embedding set. For another branch predicting trajectory distribution map, after deconvolution of each feature layer of FPN, the deconvolutional feature layers are combined with the trajectory distribution map obtained from the previous frame through channel-wise concatenation. This combined feature map is then fed to a prediction head which consists of several convolutional layers to generate the predicted trajectory distribution map of the current frame. This predicted trajectory distribution map is used to guide the model to predict the position state of the target objects. Specifically, the corrected classification result for the objects corresponding to the adaptive query embedding set is weighted by the motion information from the deconvolutional predicted trajectory distribution map; here, we call the result modified identification vector. This modified identification vector finally returns to guide the filtering of regression bounding boxes and identify these bounding boxes according to the channel index.

The tracking pipeline is illustrated in Figure 2. Except that the detection of the first frame in video sequence uses the manually initialized trajectory distribution map and the class queries obtained by iterative optimization, the detection of other frames take the trajectory distribution map and adaptive query embedding set obtained according to the detection results of the previous frame as input.

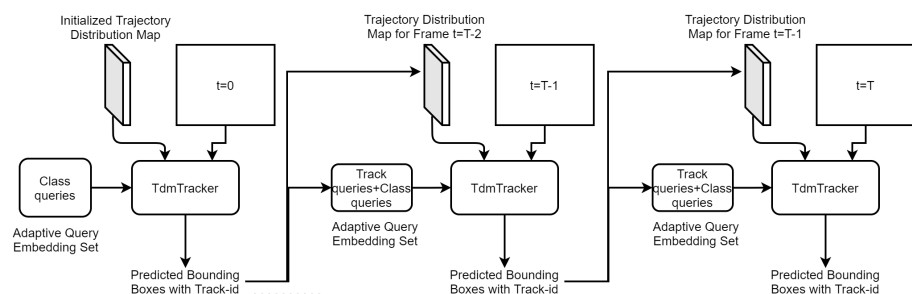

**Figure 2.** Tracking pipeline. The result of the previous frame is used to obtain the trajectory distribution map and track queries, which are inputs for current frame.

## 3.2. Model Details

### 3.2.1. Optimized Class Query

Inspired by the method of obtaining the optimal feature representation of support samples in the Siamese network of few-shot learning, we adopt iterative optimization to obtain the feature embedding of target categories. In order to generate a feature embedding which can represent the target class in multiple scales, we select several images for each category and input them into a weight shared network ResNet-34 which is the same as the backbone of TdmTracker to extract features.

The process of obtaining the optimized class query is shown in Figure 3. Image sample is first input to the ResNet-34, and three feature maps of different scales are extracted respectively. We introduce self-attention to encode the relationship between different objects and the relationship between foreground and background. Afterward, a global feature map with high-resolution and high-semantic features is generated by deconvolution

and element-wise product. Precise RoI pooling (PrRoI Pooling) [30] is adopted to extract RoI features from this global feature map. These feature maps are then channel-wise concatenated and the initial class query embedding is obtained by taking the average values over channels. In order to better distinguish target category objects from the background, we utilize $1 \times 1$ convoluted global feature map $G$ for iterative optimization.

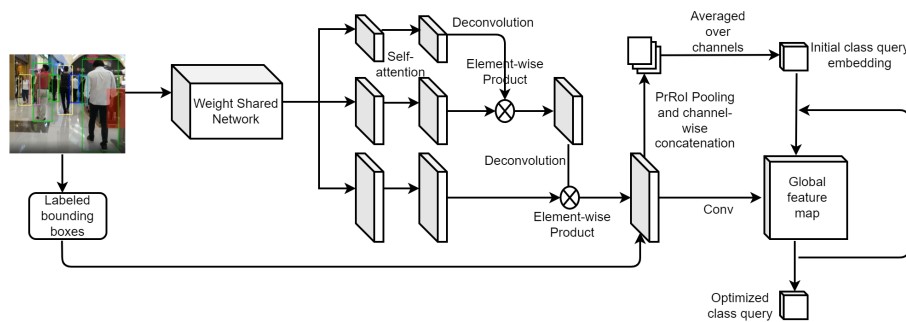

**Figure 3.** Generation of optimized class query. The combined feature map of whole image is used to extract the initial target model and iteratively optimize the query embedding.

Here are details. Suppose we express the value of position $(h, w)$ of channel $c$ in the feature map of the $i$th labeled bounding boxes as $O^i_{h,w,c} \in t^{S \times S \times C}$ and the number of labeled bounding boxes as $N$, then the initial class query embedding $Init$ is defined as

$$Init_{h,w,c} = \frac{1}{N} \sum_{i=1}^{N} O^i_{h,w,c} \tag{1}$$

Next, a cross-correlation map $Cor$ can be calculated as

$$Cor_{h,w,1} = \frac{1}{C} \sum_{i,j,c} Init_{i,j,c} \cdot G_{h+i-1,w+j-1,c}, i,j \in \{1, \dots, S\} \tag{2}$$

where the initial class query embedding $Init$ serves as a kernel to compute depth-wise cross-correlation with the global feature map $G$. Furthermore, we introduce an annotation map $T$ to compare with the cross-correlation map $Cor$ and iteratively optimize the class query embedding by reducing the deviation between $T$ and $Cor$. The annotation map is defined according to the spatial distance between the pixel and the center of the labeled bounding box. In detail, for a labeled bounding box $(\bar{x}, \bar{y}, l, w)$ with center location $(\bar{x}, \bar{y})$ and length and width $(l, w)$, the value of a pixel $(x, y)$ within this bounding box can be formulated by Gaussian distribution:

$$T_{x,y,1} = \frac{2}{lw\pi} exp\{-2[\frac{(x-\bar{x})^2}{l^2} + \frac{(y-\bar{y})^2}{w^2}]\} + v \tag{3}$$

where $v$ is used to compensate the value at bounding box center to 1. Finally, we gain the optimal class query by iteratively reducing the deviation between the cross-correlation map $Cor$ and the annotation map $T$.

3.2.2. Adaptive Query Embedding Set and Attention Feature Map

In order to update query embeddings for online tracking, we propose an adaptive query embedding set, which contains class query embeddings and adaptive track query embeddings. Our approach is to first initialize $N$ query embeddings with all elements of 0 in a query set. When the $nth$ object is detected for the first time, a uniform size feature embedding will be obtained according to the predicted bounding box of this object, used as a track query, and update the $nth$ element of adaptive query embedding set. However, if the detected object is an existing tracking object, the corresponding track query is updated.

Class queries are listed behind track queries to detect new-coming objects of the target category. In this way, the index of query embedding in the collection represents the identification of the corresponding tracked object. The adaptive query embedding set obtained from the current frame is used for the query-key attention within detection of the next frame.

We utilize attention feature map to implement query-key attention mechanism. Suppose we denote the query embedding of the *nth* tracked object as $X^n \in t^{S \times S \times C}$ and the feature map of current frame $T$ as $Y \in t^{H \times W \times C}$, the attention feature map $A$ can be formulated as

$$A_{h,w,n} = \frac{1}{C} \sum_{i,j}^{S} \sum_{c}^{C} X_{i,j,c}^n \cdot Y_{h - \frac{S-1}{2} + i, w - \frac{S-1}{2} + j, c} \tag{4}$$

where each channel of the attention feature map $A$ corresponds to a certain tracked object according to the query.

### 3.2.3. Trajectory Distribution Map

In this work, we propose a trajectory distribution map for motion prediction in order to implement tracking-conditioned detection. Unlike CenterNet [12] using predicted center offset between previous frame and current frame, our trajectory distribution map encodes all the position information from history before the current frame. Our model takes the current frame $I^{(t)} \in \mathbb{R}^{H \times W \times 3}$, the previous frame $I^{(t-1)} \in \mathbb{R}^{H \times W \times 3}$, and a trajectory distribution map $Td^{(t-1)} \in [0,1]^{\frac{H}{R} \times \frac{W}{R} \times N}$ as inputs. We initialize the trajectory distribution map as an abbreviated matrix map with a down-sample factor $R = 4$ and $N$ channels whose elements are 1, $N$ channels represent $N$ trajectories for $N$ predefined target ($N$ is greater than the number of real targets). The element of each position of all channels is initialized to 1, which means that each target to be tracked may appear at each position of the image. Afterward, each detection result will be updated to the next input trajectory distribution map. The principle of updating the trajectory distribution map is that when a new-coming object is detected in this frame, a sequential channel of the trajectory distribution map is assigned to it, and after setting all the element values of the channel to 0, the element values of this channel are calculated by a multi-dimensional Gaussian distribution of which the peak value is 1 and the standard deviation is determined by the size of the bounding box of this object. If the tracking target with existing trajectory is detected, the value is updated on the corresponding channel to make the distribution value near the detection position of the current frame greater than that of the previous detection position. To be specific, suppose we have $m$ existing trajectories, and now an object identified as index $n$ is detected in the current frame $I^{(t)}$, the channel of number $n$ in the trajectory distribution map can be expressed as

$$\widetilde{Td}_{x,y,n}^{(t)} = \begin{cases} \frac{2}{ab\pi} exp\{-2[\frac{(4x-\bar{x})^2}{a^2} + \frac{(4y-\bar{y})^2}{b^2}]\} & , \; if \; n > m \\ max(\frac{t-1}{t} \widetilde{Td}_{x,y,n}^{(t-1)}, \frac{2}{ab\pi} exp\{-2[\frac{(4x-\bar{x})^2}{a^2} + \frac{(4y-\bar{y})^2}{b^2}]\}) & , \; if \; n \leq m \end{cases} \tag{5}$$

where $(4x, 4y)$ is the corresponding coordinate on the original image and $(\bar{x}, \bar{y})$ is the center of the detected object, $a$ and $b$ are the length and width of bounding box, respectively. Since the probability value of each pixel of the trajectory distribution map obtained in this way is less than 1 by several orders of magnitude, in order to more intuitively represent the probability, we enlarge the obtained value in equal proportion $v$ as follows, so that the largest probability value is 1.

$$Td_{x,y,n}^{(t)} = v \times \widetilde{Td}_{x,y,n}^{(t)}, v = \frac{1}{max(\widetilde{Td}_n^{(t)})} \tag{6}$$

We multiply the trajectory distribution of the previous frame $Td^{(t-1)}$ by $\frac{t-1}{t}$ to represent the effect of time sequence. In this way, the closer the previous frame to the current frame,

the greater the impact on position distribution prediction. Considering that the tracker objects may leave the frame or be occluded and reappear, we decide the update strategy of the trajectory distribution map when the tracked object is not detected in the current frame as follows.

$$Td^{(t)}_{x,y,n} = \frac{t-1}{t} Td^{(t-1)}_{x,y,n}, if\ tracked\ object\ n\ is\ not\ detected \tag{7}$$

We found that our trajectory distribution map cooperates with the adaptive query set to implicitly handle short term occlusion but assigns a new identity to an object leaving the frame or being occluded for a long period. Here is an example of trajectory distribution map (see Figure 4).

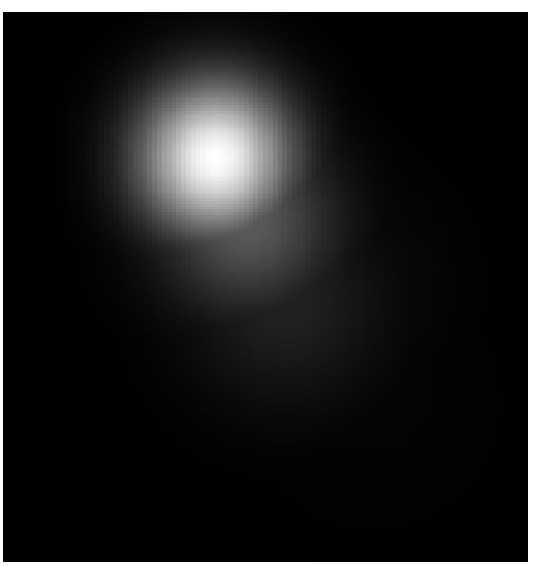

**Figure 4.** An example of trajectory distribution map. The brighter the position, the greater the probability of object distribution.

3.2.4. Matching Score Map

Referring to the practice in [19], we adopt a matching score map to transform the classification of foreground and background pixels into the classification of target and non-target category pixels. To be specific, for the feature map with the *stride* $= s$ of FPN, we denote a regression bounding box of a pixel $(x, y)$ as $B^* = (l^*, t^*, r^*, b^*)$. $l^*, t^*, r^*$, and $b^*$ are the distance from the pixel $(x, y)$ to the four sides of the regression bounding box [31]. Therefore, the coordinates of the upper left corner $(ul_x, ul_y)$ and lower right corner $(lr_x, lr_y)$ of the regression bounding box on the attention feature map $A$ can be calculated as

$$\begin{aligned} ul_x &= \frac{xs+\frac{s}{2}-l^*}{s} \quad , \quad ul_y = \frac{ys+\frac{s}{2}-t^*}{s}, \\ lr_x &= \frac{xs+\frac{s}{2}+r^*}{s} \quad , \quad lr_y = \frac{ys+\frac{s}{2}+b^*}{s}. \end{aligned} \tag{8}$$

Additionally, our matching score map *Mat* is defined as

$$Mat_{x,y,n} = \frac{1}{N_b} \sum_{i=ul_x}^{lr_x} \sum_{j=ul_y}^{lr_y} A_{i,j,n} \tag{9}$$

where $N_b$ equals $(lr_x - ul_x) \times (lr_y - ul_y)$, and the index of channels on the matching score map corresponds to the index of different query embeddings.

## 4. Training TdmTracker

### 4.1. Dataset

FSOD. FSOD [15] is a highly diverse few-shot object detection dataset. As the key of few-shot learning is the generalization ability when novel categories appear, and the detection of tracked targets in our model is based on the detection results of the previous frame, of which corresponding query embeddings can be regarded as the embeddings of new categories, we use FSOD for training the branches of classification and bounding box regression. In FSOD, there are 1000 categories with unambiguous category split for training and testing, where 531 categories are from ImageNet dataset [32] and 469 from Open Image dataset [33]. As proved in [15], this dataset with high diversity in categories is effective for learning a general rule of objects and distinguishing objects of different categories.

MOT17. MOT17 [34] is a real-world benchmark for MOT, which contains 7 training sequences and 7 test sequences. The videos were captured by stationary cameras mounted in high-density scenes with heavy occlusion. Only pedestrians are annotated and evaluated. MOT17 is the video sequence of high frame rate, of which framerate is 14–30 FPS.

### 4.2. Training Strategy and Training Loss

Since the branch of trajectory distribution map prediction in our model is only used to adjust the results of classification and bounding box regression prediction branches based on the trajectory information, and will not affect the training and inference of classification and regression prediction branches, we divide the classification and regression prediction branches and the branch of predicting trajectory distribution map into two steps for training. When we have well-trained the branch of classification and regression, we can train the branch of the predicting trajectory distribution map.

We leverage the two-way contrastive training strategy and the triplet loss of [15] to train the branch of classification and regression on FSOD so that our model can identify objects corresponding to the same query embedding from objects corresponding to different query embeddings. To be specific, we randomly choose some support images with objects of the same category as the target category in the input image $I_c$ and some support images with objects of other different categories to generate query embeddings $Q_c$ and $Q_n$, construct a training triplet $(I_c, Q_c, Q_n)$. In the image $I_c$, only the objects of category are labeled as positive, while other objects and background are labeled as negative. For this triplet, our model should not only match the same category objects between $(I_c, Q_c)$ but also distinguish objects with different classes between $(I_c, Q_n)$. Therefore, we design the training loss function as follows at the beginning:

$$L(I_c, Q_c, Q_n) = \lambda_{mc} L_{match}(I_c, Q_c) + \lambda_{mn} L_{match}(I_c, Q_n) \\ + \lambda_{l1} L_{l1}(I_c, Q_c) + \lambda_{iou} L_{iou}(I_c, Q_c) \tag{10}$$

where $L_{match}(I_c, Q_c)$ is focal loss [35] for offsetting the impact of class imbalance and makes the model pay more attention to hard examples by adjusting the weights, $L_{match}(I_c, Q_n)$ is the binary cross-entropy loss, $L_{l1}(I_c, Q_c)$ and $L_{iou}(I_c, Q_c)$ are L1 loss and generalized IoU loss [36] between normalized center coordinates and height and width of predicted boxes and ground truth box, respectively. $\lambda$ is the weighting factor for components. This multi-task loss function can be summarized as

$$L_{triplet} = \sum_i \lambda_i L_i \tag{11}$$

However, tuning these weights $\lambda_i$ by hand is a difficult and expensive process, and the resulting loss weights might be far from optimal. As suggested in [37], the optimal weighting of each task is dependent on the magnitude of the task's noise. Thus, we adopt the

automatic learning scheme for loss weights in [37]. Therefore, our loss function can be written as

$$L_{triplet} = \sum_i \frac{1}{2} \left( \frac{1}{e^{s_i}} L_i + s_i \right) \tag{12}$$

where $s_i$ is the task-dependent uncertainty for each component loss and can be modeled as learnable parameters as in [37].

After training the branch of classification and regression, we keep the well-trained parameters when training the branch of predicting trajectory distribution map on MOT17. Since each channel of our trajectory distribution map corresponds to different objects corresponding to the query embeddings, we design the total loss as a weighted sum of the loss of different channels. Because our purpose is to replace motion prediction with the predicted trajectory distribution map, we pay more attention to the frequently detected targets. Based on the fact that the fewer the number of updates in the trajectory distribution map, the greater the gap of the channel value between the previous trajectory distribution map $T - 1$ and the predicted trajectory distribution map $T$, we assign large weighted value for the loss of the channel with multiple updates and small weighted value for the loss of the channel with less updates. Furthermore, for each channel, we argue that the closer to the real position of the object, the greater the loss weight should be. Therefore, we formulate the loss for each channel as

$$L_n = \sum_{xy} Td_{xyn}(Td_{xyn} - \hat{T}d_{xyn}) \tag{13}$$

where $\hat{T}d_{xyn}$ is the predicted probability at position $(x, y)$ of the channel $n$ of the predicted trajectory distribution map. Then, the total loss of the predicted trajectory distribution map can be written as follows:

$$L_{total} = \sum_N \frac{e^{U_n}}{\sum_N e^{U_n}} L_n \tag{14}$$

where $N$ is the number of channels of the predicted trajectory distribution map and $U_n$ is the number of updates of channel $n$, that is, the number of times an object with index $n$ is detected.

### 4.3. Post-Processing

Although our TdmTracker is implemented based on the object detection model, it is obviously different from the conventional object detection model in post-processing. Since only one corresponding tracking object can be found or no corresponding object can be detected for a track query, while class queries are for detecting new-coming objects and can detect multiple different objects in the current frame, we conduct different post-processing strategies for the results obtained by track query and class query.

According to each channel of modified identification vector, we can obtain the probability of each regression bounding box belonging to the corresponding query. For each track query, we only take the bounding box with the highest probability. However, if the highest probability of a track query is lower than a predefined threshold $T_{track}$ for track queries, we believe that the track query does not detect the corresponding target. We also consider the overlap of two tracked objects. In this case, the probability value obtained by the overlapped tracked object on the pixels at the overlapping region will be much less than that of the other object. If the bounding box of the highest probability of the two tracked objects is the same one, it is considered that the bounding box belongs to the tracked object with higher probability value. For another overlapped object with lower probability value, the bounding box with the second highest probability value of pixels in the nearby $9 \times 9$ area centered on the pixel is taken as the detection bounding box.

After obtaining the bounding boxes corresponding to the track queries, we perform Soft-NMS on the remaining regression bounding boxes for all class queries together. To be specific, for each bounding box, we take the highest value of the probability belonging

to each class query as the probability that it belongs to a new-coming object. Since most objects overlap a lot with other adjacent objects in images with densely arranged objects, many correct detection results are filtered out by conventional NMS due to large IoUs. We use Soft-NMS rather than conventional NMS to process the detection results. Soft-NMS retains the correct results by reducing the lower confidence score instead of zeroing it when there is an IoU above the predefined threshold $T_{iou}$. For example, if the IoU between a bounding box $b_i$ and another bounding box $b_j$ is greater than $T_{iou}$, and $b_j$ has a higher confidence score than $b_i$, the probability score $p_i$ of $b_i$ will be recalculated according to the following formulation:

$$p_i = \left\{ \begin{array}{ll} p_i, & if\ IoU(b_i, b_j) < T_{iou}, \\ p_i(1 - IoU(b_i, b_j)), & if\ IoU(b_i, b_j) \geq T_{iou}. \end{array} \right. \tag{15}$$

Finally, we identify all bounding boxes with recalculated probabilities greater than a predefined threshold $T_{obj}$ as new-coming targets.

## 5. Experiments

### 5.1. Implementation Details and Evaluation Metrics

We implement our model based on a Pytorch framework. Our model is trained from scratch with a computer running Ubuntu 18.04 LTS. Stochastic gradient descent (SGD) is performed on Nvidia GeForce GTX 1060 with 8 GB GPU memory. The experiments utilize CUDA v10.0, cuDNN v7.5.0 to accelerate computation. Considering that too many training iterations may damage performance by making the model overfit, we take 50,000 iterations to train the branches of classification and bounding box regression on dataset FSOD, where the first 45,000 iterations are trained with learning rate 0.002 and, later, 5000 iterations are trained with learning rate 0.0002. Afterward, the branch of predicting trajectory distribution map is trained on MOT17 for 40 epochs. We use the momentum method to optimize SGD. We train the model with an initial learning rate of 0.0002, momentum of 0.9, and weight decay of 0.0005.

We adopt the standard metrics of MOT Benchmarks for evaluation, including Multiple Object Tracking Accuracy (MOTA), ID F1 Score (IDF1), Mostly tracked targets (MT), Mostly lost targets (ML), the number of False Positives (FP), the number of False Negatives (FN), and the number of Identity Switch (IDs). In addition, we use Tracking Speed in Frames Per Seconds (Hz) to measure the running speed of all methods.

### 5.2. Model Differences and Performance Comparison with Other MOT Methods on MOT17

Our proposed model TdmTracker is an end-to-end MOT method. We adopt anchor-free one-stage network to form classification and regression branches. On this basis, the adaptive query embedding set is introduced to realize attention mechanism for detecting both new-coming and tracked targets. At the same time, we propose the trajectory distribution map to introduce motion information of tracked objects, which is used to adjust the probability of different tracked objects appearing at different positions in the current frame. By adjusting the detection results by position probability, our model realizes the re-identification of targets.

On the one hand, compared with the methods based on the tracking-by-detection paradigm, such as SORT and DeepSORT, our model is more unified and lightweight, which results in a faster running speed of our model. Most tracking-by-detection methods first detect the targets through a detector, and then extract the features of different targets. When extracting features, it is very time consuming to recalculate the feature map. On the other hand, the extracted features are used to calculate the similarity of different object features in consecutive frames and combined with motion state to realize data association. Splitting the whole process into isolated subtasks may lead to local optima and much computation cost. In contrast, as an anchor-free end-to-end model, our model directly extracts the features of the whole picture, realizes feature matching through attention

mechanism, and combines motion state by weighting the predicted trajectory distribution map of the current frame. Through the identification of different tracking targets on different channels, re-identification of targets is realized. It avoids recalculating features and integrates object detection, feature extraction, and data association into a unified lightweight model. On the other hand, compared with the well-known methods based on the end-to-end paradigm, such as JDE, FairMOT, CenterTrack, CTracker, TransTrack, and TrackFormer, our model makes use of the historical position of tracked targets to improve the accuracy. JDE, FairMOT, and CenterTrack are half end-to-end models. They integrate object detection and feature extraction to avoid feature recalculation, but they also need to execute another matching algorithm to realize data association. CTracker, TransTrack, and TrackFormer are implemented as end-to-end models. Among them, CTracker integrates data association into an overall network through converting a data association problem to a pair-wise object detection problem. TransTrack and TrackFormer introduce the transformer architecture. Data association is integrated by introducing object features from the previous frame as track queries into the query-key mechanism detection process of the current frame. However, since these end-to-end models do not predict the motion state of targets and only use IoU and feature similarity for re-identification, there are ID switches when positions of targets in consecutive frames are too far away, such as blocking and high-speed motion of the object. In contrast, our model TdmTracker introduces an adaptive query embedding set to record features of all detected targets. The feature embeddings of targets will not be deleted when the targets disappear briefly in the video but continue to search all existing tracked targets. Because many targets inevitably have similar features, we utilize the trajectory distribution map to introduce historical position information of tracked objects to predict the position in the current frame, so as to improve the accuracy of position matching. In this way, our model not only searches for the corresponding target in the current frame near the position of the previous frame but also takes into account the motion of the target. It can alleviate the ID switch problem caused by occlusion and high-speed movement of objects to a certain extent.

In order to extensively evaluate our model TdmTracker, we compare it with other state-of-the-art trackers, including DeepSORT, FairMOT, CTracker, CenterTrack, TrackFormer, and TransTrack. Table 1 lists the performance comparison of our proposed model TdmTracker with other state-of-the-art MOT methods on MOT17. As the running speeds of some of these methods are not reported, for fair comparison, we re-implemented these methods on our experimental equipment and benchmark their running speeds.

**Table 1.** Performance comparison of our proposed model with other state-of-the-art MOT methods. ↓ means the smaller the better; ↑ means the larger the better. In each column, the best result is in **bold**, and the second best is <u>underlined</u>.

| Method | MOTA↑ | IDF1↑ | MT↑ | ML↓ | FP↓ | FN↓ | IDs↓ | Speed (Hz)↑ |
|---|---|---|---|---|---|---|---|---|
| DeepSORT | 60.3 | 61.2 | 31.5 | 20.3 | 36,111 | 185,301 | **2442** | 5.1 |
| FairMOT | <u>73.7</u> | **72.3** | 43.2 | 17.3 | 27,507 | 117,477 | 3303 | 6.1 |
| CTracker | 66.6 | 57.4 | 32.2 | 24.2 | <u>22,284</u> | 160,491 | 5529 | <u>9.0</u> |
| CenterTrack | 67.8 | 64.7 | 34.6 | 24.6 | **18,498** | 160,332 | <u>3039</u> | 5.2 |
| TrackFormer | 65.0 | 63.9 | <u>45.5</u> | <u>13.8</u> | 70,443 | 123,552 | 3528 | 8.7 |
| TransTrack | **74.5** | 63.9 | **46.8** | **11.3** | 28,323 | **112,137** | 3663 | 6.4 |
| TdmTracker (ours) | 70.2 | <u>65.5</u> | 43.1 | 15.4 | 30,367 | <u>115,986</u> | 3265 | **10.7** |

As Table 1 shows, our proposed TdmTracker gains competitive tracking accuracy and, meanwhile, runs faster than other state-of-the-arts. Considering the overall MOT metrics, our model achieves the second-best result at IDF1 and FN metrics, and it yields 70.2% MOTA, ranking third among all methods. In terms of tracking speed, we can clearly observe that our model has the best efficiency, which achieves the fastest speed at 10.7 Hz, outperforming other state-of-the-arts. In general, our proposed TdmTracker introduces the attention mechanism to realize the comparative learning ability similar to that of few-shot

learning and proposes the trajectory distribution map to predict the motion trajectory, which results in a comparable accuracy. At the same time, it has the fastest running speed due to our anchor-free end-to-end model architecture with a single frame input. Therefore, our model achieves a good speed-accuracy trade-off. There are some examples of tracking results produced by our proposed TdmTracker (see Figure 5).

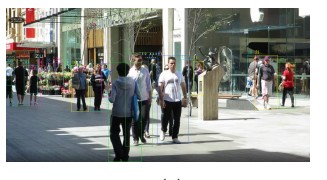 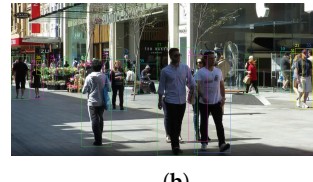 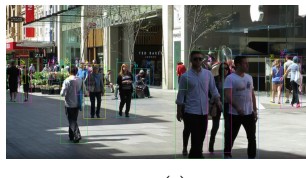

(**a**)  (**b**)  (**c**)

**Figure 5.** Some examples of tracking results produced by our proposed TdmTracker. Some tracked objects in image (**a**) are occluded in image (**b**), so that they cannot be detected, but when they reappear in image (**c**), they are identified with the ID in image (**a**), which suggests our model can handle short term occlusion.

*5.3. Ablation Studies*

In this section, we investigate the validity of different components in our proposed TdmTracker, including trajectory distribution map, optimized class query, and self-attention. Several controlled experiments on MOT17 datasets are conducted for the ablation study. To be specific, for performance analysis, we compare the following models on the MOT17 dataset to prove the effectiveness of the components in TdmTracker:

1.  **Baseline**. It only covers the classification branch and the bounding box regression branch, without guidance from the trajectory distribution map. In addition, it does not use self-attention to encode the relationship between pixel patches and the relationship between queries to obtain more representative embedding, and we use static learned object query instead of optimized class query.
2.  **Baseline + Tdm**. Based on Baseline, it adds the branch of predicting trajectory distribution map to adjust the final result.
3.  **Baseline + Tdm + Cq**. It uses optimized class query instead of static learned object query on the basis of model (2).
4.  **TdmTracker**. It uses self-attention to encode the relationship between pixel patches and the relationship between queries to obtain more representative embedding on the basis of model (3). It is the complete version of our proposed model.

The results of these models are shown in Table 2. Obviously, our trajectory distribution map, optimized class query, and introduced self-attention improve the tracking performance.

**Table 2.** Ablation study on MOT17 dataset. ↓ means the smaller the better; ↑ means the larger the better.

| Model | MOTA↑ | IDF1↑ | MT↑ | ML↓ | FP↓ | FN↓ | IDs↓ |
|---|---|---|---|---|---|---|---|
| Baseline | 52.4 | 51.9 | 30.6 | 25.4 | 64,597 | 120,895 | 7351 |
| Baseline + Tdm | 61.8 | 59.6 | 36.9 | 19.3 | 32,262 | 135,603 | 4176 |
| Baseline + Tdm + Cq | 69.1 | 64.7 | 42.6 | 15.8 | 30,980 | 115,054 | 3317 |
| TdmTracker | 70.2 | 65.5 | 43.1 | 15.4 | 30,367 | 115,986 | 3265 |

**Trajectory distribution map**. By comparing **Baseline** and **Baseline + Tdm**, we find that our proposed trajectory distribution map greatly improves MOT performance of our model. It is because that trajectory distribution map encodes the position information of tracked objects and the corresponding timing information so as to realize tracking conditioned detection and association through predicted trajectory. MOTA increases from

52.4 to 61.8 and IDF1 also increases from 51.9 to 59.6. The most significant improvement is shown in IDs, which decrease from 7351 to 4176.

**Optimized class query**. By comparing **Baseline + Tdm** and **Baseline + Tdm + Cq**, it can be found that optimized class query also plays a key role in our model. Since static learned object query focuses on whether there are objects of different sizes in different positions of the image, and is not sensitive to the category of objects, it is often confused by some unwanted objects or complex backgrounds. In contrast, optimized class query is obtained by offline training and can be distinguished from different categories of objects and complex backgrounds. Optimized class query increases the MOTA of our model by 7.3 and the IDF1 by 5.1.

**Self-attention**. As mentioned in some recent MOT works based on transformer [14,16,18], self-attention can help produce a good representation for feature by using the relationship between different parts of itself. This is confirmed in our experiment by the comparison between model **Baseline + Tdm + Cq** and **TdmTracker**. It can be observed that, except FN, other MOT metrics have a small increase after the introduction of self-attention.

### 5.4. Comparison of Different Loss Functions for Training the Branches of Classification and Regression

We first compare the discriminative ability of our model when using different loss function to train the branches of classification and regression, e.g., cross entropy loss $L_{CE}$ and our triplet loss $L_{triplet}$. For fair comparison, the models trained with both loss functions use FSOD dataset for training and are tested on MOT17 to gain results. Three aspects of performance are evaluated. Average precision (AP) is computed to evaluate the detection accuracy, and MOTA and IDs are employed to evaluate the tracking performance of the whole MOT system. Table 3 presents the results obtained from the two loss functions with different loss weighting strategies. According to Table 3, two observations can be made.

**Table 3.** Comparing different losses with different loss weighting strategies. App.Opt means using a set of approximate optimal loss weights obtained by cross validation and uncertainty means using task-dependent uncertainty to obtain loss weights as described in Section 4.2. ↓ means the smaller the better; ↑ means the larger the better. In each column, the best result is in **bold**.

| Loss Function | Weighting Strategy | Detection | MOT | |
|---|---|---|---|---|
| | | AP↑ | MOTA↑ | IDs↓ |
| $L_{CE}$ | App.Opt | 78.4 | 60.7 | 4266 |
| $L_{CE}$ | Uncertainty | 79.5 | 62.2 | 4037 |
| $L_{triplet}$ | App.Opt | 83.6 | 68.3 | 3603 |
| $L_{triplet}$ | Uncertainty | **84.8** | **70.2** | **3265** |

First, we can clearly find that no matter which weighting strategy is used, the result of using $L_{triplet}$ for training is better than that of using $L_{CE}$ for training. We analyze that this is because using $L_{triplet}$ can make better use of the advantages of the FSOD dataset, that is, there are many categories and less samples in each category. Moreover, we believe that using $L_{triplet}$ for training makes the model more inclined to remember the method of comparing different categories, while using $L_{CE}$ for training makes the model more inclined to learn the representation of the category. Therefore, when a new category appears or the appearance of an object changes greatly, the model trained with $L_{triplet}$ is more adaptive.

Second, it can be seen that correctly weighting loss terms is of paramount importance for multi-task learning. For each loss function, using task-dependent uncertainty to obtain loss weights is better than using a set of approximate optimal loss weights obtained by cross validation. This is because finding the approximate optimal loss weights by cross validation is a difficult and expensive process, and the final result is often far from optimal. However, as far as using task-dependent uncertainty, optimizing the loss weights using a homoscedastic noise term allows for the weights to be dynamic during training and the uncertainty term decreases during training which improves the optimization process [37].

## 6. Conclusions

The research on MOT detection is of great significance. In this work, we propose a novel anchor-free attention mechanism-based end-to-end MOT model TdmTracker, which introduces the trajectory distribution map to guide position prediction and uses the adaptive query embedding set and query-key attention mechanism to detect tracked targets in the current frame. Our work presents a novel unified and lightweight architecture of the MOT model. Moreover, we utilize optimized class query instead of static learned object query to detect new-coming objects, which realizes the specialization of the tracking target category in the tracking process. Furthermore, the model uses a comparative learning mode similar to that in few-shot learning so that when there are novel category targets to track, only the optimized class query of the novel category needs to be trained without retraining the model. The experimental result on the MOT17 dataset shows that our proposed TdmTracker has comparable tracking accuracy with other state-of-the-art MOT methods and achieves the highest tracking running speed. Such good time-accuracy trade-off of our model makes it possible to be applied on applications with real-time requirements. In addition, our model uses the features extracted from the deep network rather than handcrafted features; thus, it has good generality in application objects. However, our model has a limitation on the number of tracking targets in the application scenario. Unlike other state-of-the-art models, which delete features for targets that have not appeared for a period of time, when the number of targets to be tracked is large, the adaptive query embedding set of our model needs to set a large number of channels to store the features of each target. At the same time, the trajectory distribution map also needs the same number of channels to predict the motion state of each tracked target. A lot of calculation will slow down the model. Moreover, when dealing with the cases in which most targets in the video only appear for a short time, it will obviously waste much computation cost. Therefore, our model expects the duration of tracking targets to account for most of the total tracking time, and the number of possible tracking targets is small. In the future, we believe that our model can be used in the research of UAV tracking and the tracking of suspicious people in some important places.

**Author Contributions:** Formal analysis, Y.G.; investigation, Y.G.; methodology, Y.G. and Y.H.; supervision, X.G., R.H., Q.G. and Y.H.; writing—original draft, Y.G.; writing—review and editing, Y.G. All authors have read and agreed to the published version of the manuscript.

**Funding:** This research was funded by the National Natural Science Foundation of China (51805264).

**Institutional Review Board Statement:** Not applicable.

**Informed Consent Statement:** Not applicable.

**Conflicts of Interest:** The authors declare no conflict of interest.

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
