# Peer review of "TdmTracker: Multi-Object Tracker Guided by Trajectory Distribution Map"

_electronics, doi:10.3390/electronics11071010_

Round 1
Reviewer 1 Report
This paper proposed an anchor-free attention mechanism based end-to-end MOT model called TdmTracker. This model introduced trajectory distribution map to guide position prediction, and used adaptive query embedding set and query-key attention mechanism to detect tracked targets. The result is interesting. However, it seems that the overall review to the state-of-art models are not sufficiently provided in the introduction section and therefore the significance of the present study is not well elucidated. For instance, as far as this reviewer understands, mature models exists in detecting objects in figures, such as GoogLeNet, ResNet (https://arxiv.org/abs/1512.03385), etc. What are the difference between this model and the well-known published one? The author claims there were improvements from the model in terms of significance, accuracy, predicting speed, etc. This reviewer suggests to compare the model performance with the mature one so that the importance could be further revealed.
Reviewer 2 Report
In this article, the authors have presented interesting work. I am appreciating this work. This article can be considered for publication after the incorporation of suggestions and observations as mentioned below. I would like to see this article after major revision. There are several major concerns: 1) The paper is generally well written, but it would definitely benefit from proof reading by a native speaker. There are many grammatical mistakes and informal statements that should be avoided in a scientific text. 2) The organization of the work is a bit disordered should also be compared against previous work. 3) Latest work is missing and overlooked. So, the authors should add the suggested relevant work like: - * Gait Recognition Based on Vision Systems: A Systematic Survey, Journal of Visual Communication and Image Representation, 103052 * An efficient content based image retrieval system using BayesNet and K-NN, Multimedia Tools and Applications 77 (16), 21557-21570 * Underwater image enhancement using blending of CLAHE and percentile methodologies, Multimedia Tools and Applications 77 (20), 26545-26561 * Content-based image retrieval system using ORB and SIFT features, Neural Computing and Applications 32 (7), 2725-2733 * 2D-human face recognition using SIFT and SURF descriptors of face's feature regions, The Visual Computer, 1-10 * Improved object recognition results using SIFT and ORB feature detector, Multimedia Tools and Applications 78 (23), 34157-34171 4) The contribution should also be presented in the context of the state of the art in making the models lightweight. 5) The authors should include some discussion on whether the proposed technique could be made more generic to be used in other networks designed for solving other tasks, or whether it is limited only to the problem of estimating human pose. 6) Please discuss more about drawbacks, any future research opportunities.
Round 2
Reviewer 1 Report
This reviewer is satisfied with the author's response.
Reviewer 2 Report
Accepted